# Bioactivity-Guided Fractionation and NMR-Based Identification of the Immunomodulatory Isoflavone from the Roots of *Uraria crinita* (L.) Desv. ex DC

**DOI:** 10.3390/foods8110543

**Published:** 2019-11-03

**Authors:** Ping-Chen Tu, Chih-Ju Chan, Yi-Chen Liu, Yueh-Hsiung Kuo, Ming-Kuem Lin, Meng-Shiou Lee

**Affiliations:** 1Program for Cancer Biology and Drug Discovery, China Medical University and Academia Sinica, Taichung 404, Taiwan; pingchen.tu@gmail.com; 2Department of Chinese Pharmaceutical Sciences and Chinese Medicine Resources, China Medical University, Taichung 404, Taiwan; ruby24301@gmail.com (C.-J.C.); kuoyh@mail.cmu.edu.tw (Y.-H.K.); 3Institute of Biomedical Science and Rong Hsing Research Center for Translational Medicine, National Chung-Hsing University, Taichung 402, Taiwan; sealioler@gmail.com; 4Department of Biotechnology, Asia University, Taichung 413, Taiwan; 5Chinese Medicine Research Center, China Medical University, Taichung 404, Taiwan

**Keywords:** *Uraria crinita*, isoflavone, genistein, NMR-based identification, dendritic cells

## Abstract

*Uraria crinita* is used as a functional food ingredient. Little is known about the association between its immunomodulatory activity and its metabolites. We applied a precise strategy for screening metabolites using immunomodulatory fractions from a *U. crinata* root methanolic extract (UCME) in combination with bioactivity-guided fractionation and NMR-based identification. The fractions from UCME were evaluated in terms of their inhibitory activity against the production of pro-inflammatory cytokines (IL-6 and TNF-*α*) by lipopolysaccharide (LPS)-stimulated mouse bone marrow-derived dendritic cells (BMDC). The role of the isoflavone genistein was indicated by the ^1^H NMR profiling of immunomodulatory subfractions (D-4 and D-5) and supported by the result that genistein-knockout subfractions (D-4 *w*/*o* and D-5 *w*/*o*) had a lower inhibitory activity compared to genistein-containing subfractions. This study suggests that genistein contributes to the immunomodulatory activity of UCME and will help in the standardization of functional food.

## 1. Introduction

*Uraria crinita* (L.) Desv. ex DC. (UC) belongs to the family of Leguminosae. It is a popular and commercially important medicinal plant, distributed widely in Taiwan and cultivated mostly in Mingjian Township, Nantou (approximately 20–60 hectares per year). UC roots, also known as “Taiwanese ginseng” due to the similar potency and aroma of its decoction and ginseng, have been traditionally used to coordinate the gastrointestinal system, thanks to their detumescent and antipyretic effects, indicating immunomodulatory activity [1]. UC roots are used as dietary supplements for treating childhood skeletal dysplasia. UC roots have therefore been developed as valuable and commercial functional food in Taiwan. This herb has been shown to have antioxidant [2] and antidiabetic activities [3] and the potential to stimulate bone formation and regeneration [4]. Previous phytochemical investigations on UC roots led to the isolation of fatty acids, steroids, triterpenoids, phenolics, lignans, flavonoids, and isoflavonoids [2,4,5,6]. However, little is known about the association between the immunomodulatory activity and the metabolites in this herb.

Dendritic cells (DCs), acting as antigen-presenting cells (APCs), are the major leukocytes, with a critical role in regulating adaptive immune responses [7]. Immature DCs, characterized by a high antigen uptake ability and poor antigen-presenting function, reside in the peripheral tissues, where they regularly uptake and process self-antigens and maintain self-tolerance [7]. Upon activation, immature DCs undergo maturation and migrate to adjacent lymph nodes or to the lymph organs, after the recognition of pathogen-associated molecular patterns and damage-associated molecular patterns by pattern recognition receptors, mostly Toll-like receptors (TLRs) [8]. This process is accompanied by the upregulation of the expression of major histocompatibility complex (MHC) class II molecules and several co-stimulatory molecules (CD40, CD80, and CD86) on the surface of cells [9]. Mature DCs generate more pro-inflammatory cytokines (TNF-*α*, IL-6, and IL-12) required for T cell activation and have the ability to present antigens to T cells, linking the innate and adaptive immune systems [9]. Therefore, targeting DCs is a promising strategy for immunomodulation. Medicinal herbs, which modulate the function of DCs, can potentially be developed into botanical drugs for treating immune disorders.

The promising potential of the herbal industry can only be achieved through the standardization of the composition of herbal products and the assurance of proper quality control [10]. However, the variable constituents of herbal products, owing to genetic, cultural, and environmental factors, have made the quality of herbal medicines difficult to control. Nuclear magnetic resonance (NMR) has been demonstrated as one of the best analytical approaches to identifying metabolites and providing both qualitative and quantitative information [11]. While NMR suffers from a relatively poor sensitivity compared to mass spectrometry (MS), it has some unique advantages over MS-based approaches, including a non-destructive nature, high robustness, high reproducibility, high reliability, and powerful ability to provide structural information for unknown metabolites [12]. NMR spectroscopy could offer structural elucidation, achieved by the chemical shift, multiplicity, coupling constant, and integration of (primary or secondary) metabolite signals in crude extracts. Additionally, the ^1^H NMR signal intensity, proportional to the relative number of protons, could provide useful information about the quantity of the different metabolites in herbal extracts. Therefore, NMR has been successfully used for fingerprinting and the metabolite discrimination of herbs [13,14]. Currently, NMR-based metabolomic analysis is widely used in studies of nutrients, environment, plant physiology, drug metabolism, toxicology, as well as for diagnoses and for the quality control of herbal products [10,15,16].

In this study, a combination of bioactivity-guided fractionation and NMR-based identification led to the elucidation of the central role of the immunomodulatory isoflavone genistein present in UC root methanolic extract (UCME) against the activation and maturation of lipopolysaccharide (LPS)-stimulated DCs. This strategy could prevent unnecessary time-consuming isolation procedures and provide a rapid tool for the identification of the active ingredients of herbs. Our findings suggest that UC roots can be applied as an immunosuppressive functional food, of which genistein can be a chemical marker for quality control. The standardization of UC roots could therefore be improved.

## 2. Results and Discussion

### 2.1. The EtOAc-Soluble Fraction from UCME Inhibited LPS-Stimulated DC Activation

The immunomodulatory effects of UC roots on DCs have not been reported. Here, the Gram-negative bacterial endotoxin LPS was used to stimulate bone marrow-derived dendritic cell (BMDC) activation, as a model for investigating the immunomodulatory effects of UC roots on DCs. First, the immunomodulatory effects of UCME and its EtOAc-, *n*-BuOH-, and H_2_O-soluble fractions were evaluated against the production of pro-inflammatory cytokines, including TNF-*α* and IL-6, which is a hallmark of DC activation. As shown in Figure 1A, LPS-stimulated BMDC activation was suppressed by UCME, and UCME ability to inhibit DC activation was mainly associated with its EtOAc-soluble fraction. In addition, the treatment with UCME and its various partitioned fractions, at concentrations below 100 μg/mL, did not exhibit any cytotoxicity in BMDC (data not shown). In summary, our results revealed that the EtOAc-soluble fraction of UCME may contain immunomodulatory phytochemicals which attenuate the activity of DCs.

### 2.2. Bioactivity-Guided Fractionation and NMR-Based Identification of the EtOAc-Soluble Fraction of UCME

The EtOAc-soluble fraction of UCME was subjected to silica gel column chromatography (EtOAc/*n*-hexane/MeOH, gradient), and each collected fraction was analyzed by thin-layer chromatography (TLC) and assign to one of nine main fractions (Fr. A to I, Figure 2).

Among them, fraction D significantly inhibited the production of TNF-*α* and IL-6 in BMDC (Figure 1B). Furthermore, the subfractions D-4 and D-5 from fraction D indicated the most potent inhibitory effects against DC activation (Figure 1C).

In order to elucidate the association between bioactivity and metabolites in subfractions D-1 to D-6, ^1^H NMR spectroscopy was conducted (Figure 3). This could offer structural elucidation, achieved by the chemical shift, multiplicity, coupling constant, and integration of metabolite signals in the mixtures.

According to the ^1^H NMR profiling of subfractions D-1 to D-6, the characteristic singlet signals (*δ*_H_ 8.13) indicated the presence of isoflavonoids in D-3, D-4, D-5, and D-6. The inhibitory activity of subfractions D-4 and D-5 was then related to the additional signals [*δ*_H_ 7.44 (2H, d, *J* = 8.7 Hz), 6.88–6.94 (overlaps), 6.40 (1H, d, *J* = 2.2 Hz), and 6.28 (1H, d, *J* = 2.2 Hz)], which were not visible for subfractions D-3 and D-6. A pair of aromatic protons with a *meta* coupling (*J* = 2.2 Hz) indicated the presence of a 1,3,4,5-tetrasubstituted aromatic ring. An aromatic proton signal at *δ*_H_ 7.44 (2H, d, *J* = 8.7 Hz) revealed that the other proton signal might be overlapping at *δ*_H_ 6.88–6.94 ppm, suggestive of a 1,4-disubstituted aromatic ring. To determine the overlapping peaks at *δ*_H_ 6.88–6.94 ppm, the ^13^C NMR and 2D NMR experiments (Appendix A), including heteronuclear single-quantum correlation (HSQC) and heteronuclear multiple-bond correlation (HMBC), were then conducted. On the basis of these results, genistein was identified [17] from the genistein-containing subfractions (D-4 and D-5). Bioactive fractions, together with the active ingredient, were rapidly obtained from the combination of bioactivity-guided fractionation and NMR-based identification of the UCME EtOAc-soluble fraction.

### 2.3. The Role of Genistein in Modulating LPS-Stimulated DC Activation of UCME

To clarify the role of genistein in modulating LPS-stimulated DC activation, the genistein-containing subfractions D-4 and D-5 were further isolated by normal-phase semipreparative HPLC to isolate genistein (24 mg/1.136 kg dry material) and genistein-knockout subfractions (D-4 *w*/*o* and D-5 *w*/*o*, Appendix A). The yield of genistein was 15 times higher than that achieved in a previous study [6]. These subfractions were then evaluated in terms of their inhibitory activity by measuring cytokine production in LPS-stimulated DCs. As shown in Figure 4A, the genistein-containing subfractions (D-4 and D-5) significantly suppressed the production of cytokines (IL-6 and TNF-*α*), while the genistein-knockout subfractions D-4 *w*/*o* and D-5 *w*/*o* showed lower inhibitory activity. As illustrated in Figure 4B, concerning TNF-*α* production, the inhibition percentage of the genistein-knockout subfraction D-4 *w*/*o* (9% at 25 μg/mL and 18% at 50 μg/mL) was dramatically lower than that of the genistein-containing subfraction D-4 (35% at 25 μg/mL and 55% at 50 μg/mL). 

In addition to the identification of genistein, the fractionation and HPLC isolation of UCME identified eight compounds, including an isoflavone (lupinalbin A [18]), three phenolic acids (*p*-hydroxybenzoic acid [19], salicylic acid [20], and vanillic acid [19]), two fatty acids (monomethyl succinate [21] and 1,10-decanedioic acid [22]), and two steroids (a mixture of *β*-sitosterol and stigmasterol [23]). Their structures (Figure 5) were identified by comparing their spectroscopic data with data in the literature. Among the isolates, lupinalbin A, *p*-hydroxybenzoic acid, vanillic acid, monomethyl succinate, and monomethyl succinate were isolated from this herb for the first time. All isolates were also assessed in terms of their inhibitory effect on DC activation. However, only the analogue of genistein, lupinalbin A, showed a moderate inhibitory activity against LPS-stimulated DC activation (Appendix A). 

^1^H NMR signal intensity is absolutely proportional to the relative number of protons. Therefore, the relative quantity of the different metabolites could easily be observed in the ^1^H NMR spectra (Figure 3). The major metabolites (identified by signals with a stronger intensity) in subfractions D-4 and D-5 are shown in Table 1. Interestingly, the ^1^H NMR profiling of subfractions D-4 and D-5 revealed the presence of other unknown genistein derivatives, which might contribute slightly to the immunomodulatory effects, because of their low abundance or poor activity. Additionally, the well-known bioactive isoflavonoids, including daidzein, formononetin, equol, and glycitein, were not present as major metabolites in fraction D. Therefore, genistein was suggested as having the central role in the modulation by UCME of LPS-stimulated DC activation.

### 2.4. LPS-Stimulated DC Maturation was Impaired by Genistein at Non-Cytotoxic Concentrations

To exclude the possibility that genistein caused cytotoxicity, the cell viability of DCs was determined via the CCK8 assay. As shown in Figure 6, genistein induced a significant level of DC death at 40 μM. Thus, the significant inhibition of cytokine production by 40 μM genistein should be attributed to its cytotoxicity in DCs. However, the LPS-stimulated production of pro-inflammatory cytokines (TNF-*α*, IL-6, and IL-12) was suppressed by genistein below 20 μM, suggesting that genistein possesses an immunosuppressive activity. Therefore, we selected concentrations below 20 μM of genistein for further assessment of DC maturation. The complex process of DC maturation is accompanied by the upregulation of the expression of MHC class II molecules and three major co-stimulatory molecules (CD40, CD80, and CD86) on the surface of DCs. As shown in Figure 7, LPS stimulation upregulated the expression of MHC class II and also of the co-stimulatory molecules (CD40, CD80, and CD86) in DCs, while genistein treatment significantly decreased the expression levels of all these molecules. These data indicated that genistein indeed impaired LPS-stimulated DC maturation at non-cytotoxic concentrations.

Isoflavones, including genistein, daidzein, and glycitein, are generally found in leguminous plants and have been reported as antioxidants and immunosuppressant agents, capable of suppressing the allergic sensitization to peanuts by regulating human monocyte-derived dendritic cell function [24]. The majority of the dietary isoflavonoids are present in inactive glycosides forms (e.g., genistin) and then converted to active aglycone forms (e.g., genistein) by the bacterial microbiote in the digestive tract. Thus, DCs have direct access to dietary antigens and are therefore poised to uptake isoflavones directly from the lumen [24]. Previously, genistein was shown to have promising activities, such as neuroprotective effects by improving hippocampus neuronal cell viability and proliferation in vitro [25], antioxidant capacity by regulating *β*-oxidation and energy metabolism in vivo [26], and anti-inflammatory effects by inhibiting the ERK pathway [27] and NF-*κ*B-dependent gene expression in TLR4-stimulated DCs [28].

In this study, the association between genistein and the immunomodulatory effect of UCME was carefully elucidated through the combination of bioactivity-guided fractionation and NMR-based identification. An ^1^H NMR-based metabolomics approach was applied to partially purified subfractions D-1 to D-6 from UCME. ^1^H NMR profiling suggested the presence of genistein in the D-4 and D-5 subfractions, which exhibited a stronger inhibitory activity against cytokine production in LPS-stimulated DCs. Genistein was therefore supposed to be a possible marker of the immunosuppressive activity of UCME. This suggestion was supported by the results of the genistein-knockout subfractions, which showed a much lower inhibitory activity. Moreover, HPLC isolation of the D-4 and D-5 subfractions provided other compounds with poor inhibitory activity. On the basis of this evidence, genistein could be used as a chemical marker for the quality control of the potentially immunosuppressive functional food UC roots. A literature survey disclosed that the ^1^H NMR-based metabolomics approach for screening bioactive secondary metabolites has not been widely used [29,30]. Our study provides a powerful tool for discovering the active ingredients in immunosuppressive functional foods. This approach can be applied for investigating the active ingredients of herbal products.

## 3. Materials and Methods

### 3.1. General Experimental Procedures

Column chromatography (CC) was performed on Silica gel 60 (40–63 µm, Merck, Darmstadt, Germany). Thin-layer chromatography (TLC) was performed on silica gel 60 F254 plates (200 µm, Merck). High-performance liquid chromatography (HPLC) was performed, using Keystone Spherisorb silica (5 µm, 250 × 10 mm), on a Knauer Smartline 2400 refractive index (RI) detector and a Knauer Smartline 100 pump. The NMR experiments were performed on a Bruker DRX-500 NMR spectrometer (Bruker, Rheinstetten, Germany). Flow cytometry was conducted using a BD FACSCanto II Flow Cytometer (BD Biosciences, CA, USA).

### 3.2. Sample Preparation and Isolation

UC was purchased from Mingjian Township, Nantou, Taiwan. The procedure of extraction and isolation is summarized in Figure 2. Briefly, the roots (1.136 kg) were pulverized into a fine powder and extracted with methanol (12 L). The supernatant was collected and concentrated under reduced pressure to obtain the methanolic extract (UCME, 80 g). A portion of the residue (60 g) was suspended in H_2_O and sequentially fractionated with EtOAc and *n*-BuOH. The EtOAc-soluble fraction (9.6 g) was then subjected to silica-gel column chromatography (150 g, 70–230 mesh), using a gradient solvent system of *n*-hexane, EtOAc, and MeOH as a mobile phase. Each fraction, from which a sample was collected for the immunomodulating assessment, was analyzed by thin-layer chromatography (TLC) and assigned to one of 9 main fractions (Fr.A to Fr.I). A mixture of *β*-sitosterol and stigmasterol was obtained from fraction C (*n*-hexane/EtOAc = 9/1). The subfractions of fraction D (*n*-hexane/EtOAc = 7/3) were further analyzed by ^1^H NMR, indicating the presence of genistein in the D-4 and D-5 subfractions. The genistein-knockout subfractions (D-4 *w*/*o* and D-5 *w*/*o*) were obtained by HPLC and evaluated in terms of their immunomodulating activity. 

### 3.3. HPLC Conditions Used for the D-4 and D-5 Subfractions

D-4 and D-5 were isolated by semipreparative HPLC (dichloromethane/acetone = 85/15, flow rate = 3 mL/min) to obtain the genistein-knockout subfractions D-4 *w*/*o*, D-5 *w*/*o* and genistein (24.0 mg, t_R_ = 8.0 min). The genistein-knockout subfractions D-4 *w*/*o* and D-5 *w*/*o* were further isolated by semipreparative HPLC (*n*-hexane/acetone = 2/1, flowrate = 3 mL/min) to obtain salicylic acid (t_R_ = 8.3 min), lupinalbin A (t_R_ = 11.0 min), 1,10-decanedioic acid (t_R_ = 11.3 min), monomethyl succinate (t_R_ = 13.2 min), *p*-hydroxybenzoic acid (t_R_ = 16.7 min), and vanillic acid (t_R_ = 18.5 min).

### 3.4. NMR Analysis

The samples were dissolved in the deuterated solvent acetone-*d*_6_ and put into a 5 mm NMR tube. All experiments were performed on a Bruker DRX-500 NMR spectrometer (Bruker, Rheinstetten, Germany), operating at a frequency of 500 MHz for ^1^H NMR observation, and 125 MHz for ^13^C NMR observation (at room temperature). The 2D NMR experiments included heteronuclear single-quantum correlation (HSQC) and heteronuclear multiple-bond correlation (HMBC). NMR spectra were carefully processed with the TOPSPIN2.1® software (Bruker). The spectra recorded in acetone-*d*_6_ were referenced to the solvent signal at *δ*_H_ 2.05 ppm and *δ*_C_ 29.92 ppm.

### 3.5. Preparation of BMDC

The ICR mice used in this study were obtained from the National Laboratory Animal Center (NLAC, Taipei, Taiwan). The mouse bone marrow-derived DCs were prepared as described previously [31]. Bone marrow cells were isolated from tibias and femurs and then seeded on 6-well plates (Corning) in 4 mL/well RPMI 1640 medium (Thermo), with 10% FBS and 10 ng/mL recombinant mouse GM-CSF and IL-4 (Peprotech). On day 3 and 5, a 2 mL/well fresh medium containing 10 ng/mL GM-CSF and IL-4 was added. On day 7, BMDCs (> 80% CD11c^+^ cells) were harvested and used for all experiments. 

### 3.6. Measurement of Cytokine Production

Cytokine production was measured using an enzyme-linked immunosorbent assay (ELISA), as described previously [31]. The DCs were treated with 100 ng/mL lipopolysaccharide (LPS) (Sigma) or LPS + sample for 6 h for TNF-*α*, IL-6, and IL-12p70 determination. The production of cytokines was measured using the ELISA kit (eBioscience).

### 3.7. Cytotoxicity Assessment

DCs were treated with genistein (2.5, 5, 10, 20, and 40 μM) for 24 h. The cells were then measured in terms of their cell viability by the CCK-8 assay (Sigma), according to standard protocols, as described previously [32]. Triplicate treatments were performed for each sample in all experiments.

### 3.8. Analysis of DC Maturation

Maturation was determined by measuring the upregulation of MHC class II and three co-stimulatory molecules (CD40, CD80, and CD86), as described previously [31,32]. DCs were untreated or treated with LPS (100 ng/mL) or LPS + genistein (5, 10, and 20 μM) for 24 h. Cell aggregation was examined by microscopy (40×). Then, the cells were stained with monoclonal antibodies (mAbs), specific to mouse CD11c, MHC class II, CD40, CD80, and CD86 (Biolegend), and analyzed by flow cytometry. The fluorescence intensity of MHC class II, CD40, CD80, and CD86 was determined, following gating with a forward side scatter (FSC) and CD11c^+^ expression. The change in the mean fluorescence intensity (MFI) from LPS alone to LPS + genistein was indicated.

### 3.9. Data Analysis

The significance of the suppressions was determined using one-way ANOVA, followed by Scheffe’s test. A value of * *p* < 0.05 was considered significant. Values of ** *p* < 0.01 and *** *p* < 0.001 were considered highly significant.

## 4. Conclusions

In the present study, we assessed the effect of UC roots on the immune function of DCs and found that the immunomodulatory effect of UCME was mainly associated with its EtOAc-soluble fraction. After one-step chromatography, genistein was rapidly identified by the ^1^H NMR profiling of the immunomodulatory subfractions (D-4 and D-5). The central role of genistein in the immunomodulatory activity of UC roots was supported by the result that the genistein-knockout subfractions (D-4 *w*/*o* and D-5 *w*/*o*) had a lower inhibitory activity.

Importantly, this work elucidates a rapid strategy for identifying immunomodulatory phytochemicals, distinguishing the chemical marker(s) for quality control and providing a rationale for the traditional immunomodulatory use of UC roots. The findings indicated that UC roots can potentially be used as an immunosuppressive functional food, of which genistein can be a chemical marker for quality control. In conclusion, our strategy will help in the standardization of *U. crinita* roots, a famous Taiwanese functional food.

## Figures and Tables

**Figure 1 foods-08-00543-f001:**
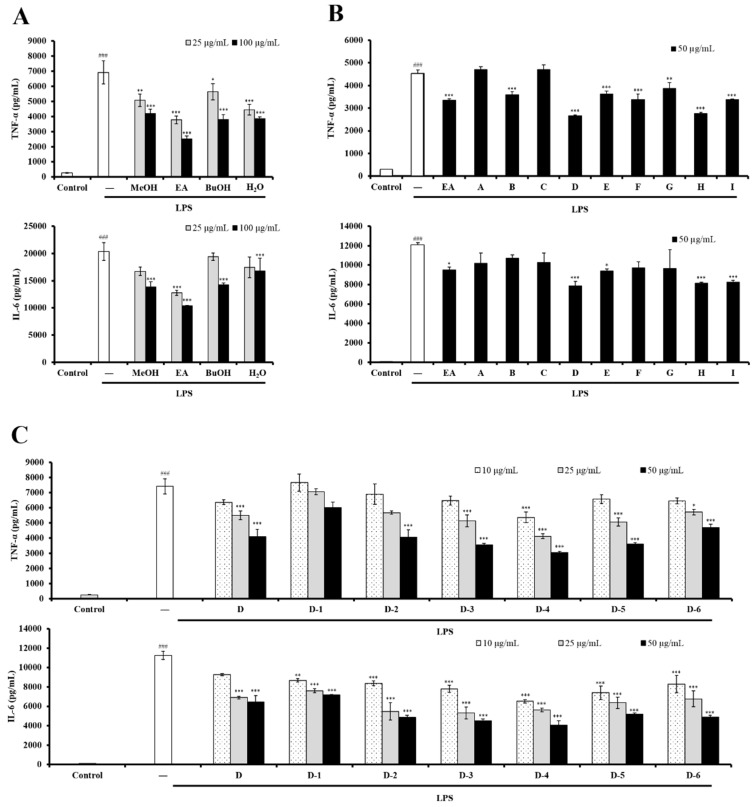
The effects of UC methanolic extract and of its EtOAc-, *n*-BuOH-, H_2_O-soluble fractions and subfractions of the EtOAc-soluble fraction on pro-inflammatory cytokine production in LPS-stimulated dendritic cells (DCs). DCs were untreated or treated with LPS (100 ng/mL, white bar). (**A**) Methanolic extract and EtOAc, *n*-BuOH, and H_2_O subfractions (25 μg/mL, gray bar, or 100 μg/mL, black bar) were used. (**B**) EtOAc and Fr. A to I subfractions (50 μg/mL, black bar) were used. (**C**) Fr. D and D-1 to D-6 subfractions (10 μg/mL, hatch bar, 25 μg/mL, gray bar, or 50 μg/mL, black bar) were used. Supernatants were collected 6 h after the treatment. The production of cytokines (TNF-*α* and IL-6) was measured by ELISA. The data shown are the mean ± SD of three independent experiments; ^###^
*p* < 0.001; * *p* < 0.05; ** *p* < 0.01; *** *p* < 0.001 (Scheffe’s test) for comparisons of the treated and untreated LPS-stimulated DC samples.

**Figure 2 foods-08-00543-f002:**
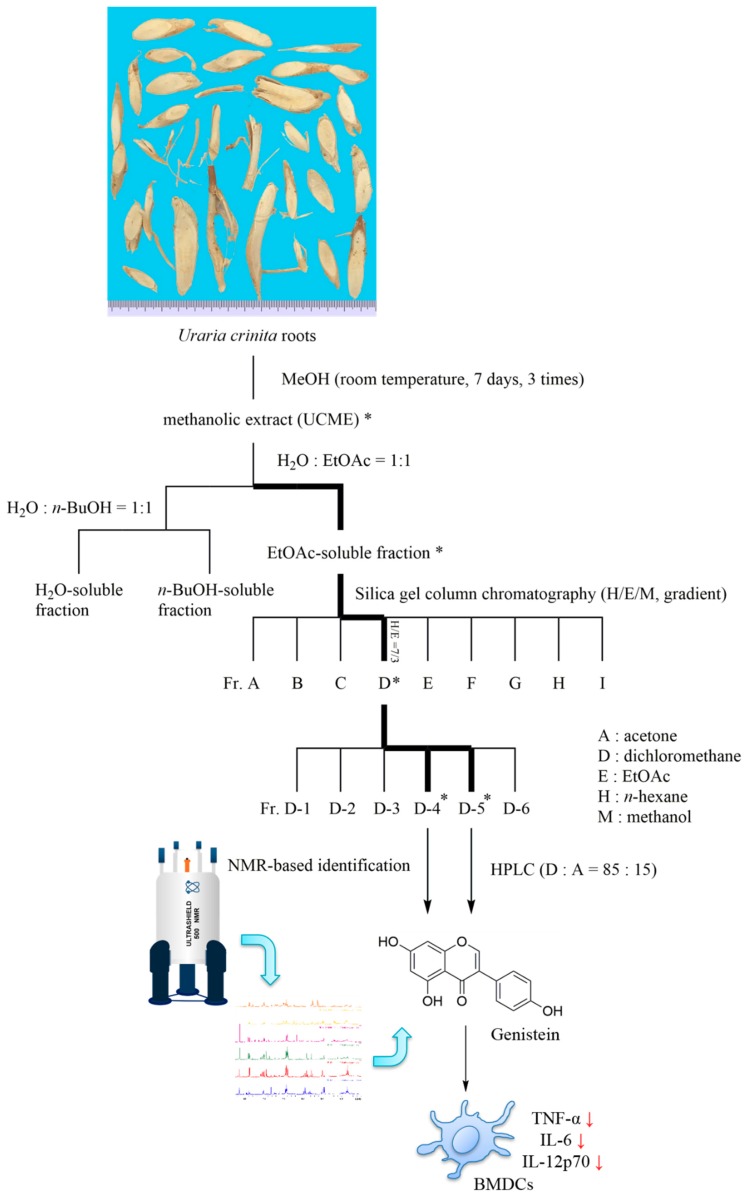
Bioactivity-guided fractionation and NMR-based identification of genistein from the roots of *Uraria crinita* (UC). * indicates the most potent subfractions or constituents against pro-inflammatory cytokine production in lipopolysaccharide (LPS)-stimulated DCs. UCME: UC root methanolic extract, BMDCs: bone marrow-derived dendritic cells.

**Figure 3 foods-08-00543-f003:**
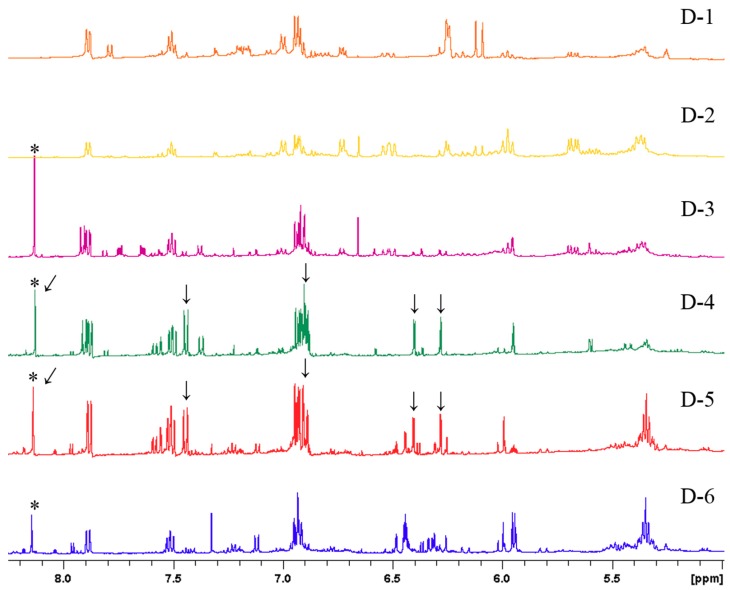
Selected ^1^H NMR profiling (acetone-*d*_6_, 500 MHz) of subfractions D-1 to D-6. * indicates the characteristic singlet signals, *δ*_H_ 8.13, of isoflavones; ↓ indicates the ^1^H spectral data of genistein: *δ*_H_ 7.44 (2H, d, *J* = 8.7 Hz), 6.88–6.94 (overlaps), 6.40 (1H, d, *J* = 2.2 Hz), and 6.28 (1H, d, *J* = 2.2 Hz).

**Figure 4 foods-08-00543-f004:**
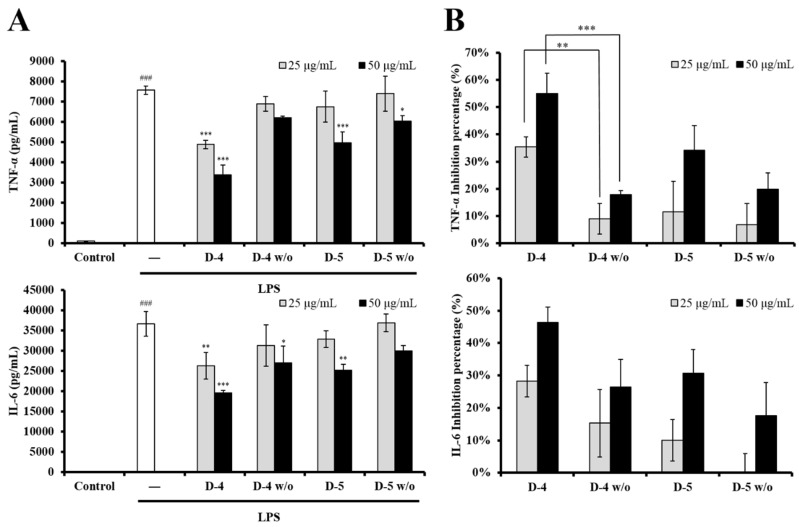
The effects of the subfractions, containing or not containing genistein, on the production of pro-inflammatory cytokines in LPS-stimulated DCs. (**A**) DCs were untreated or treated with LPS (100 ng/mL, white bar), LPS + genistein-containing subfractions, or LPS + genistein-knockout subfractions (25 μg/mL, gray bar or 50 μg/mL, black bar), as indicated. Supernatants were collected 6 h after the treatment. The production of cytokines (TNF-*α* and IL-6) was measured by ELISA. The data shown are the mean ± SD of three independent experiments. ^###^
*p* < 0.001; * *p* < 0.05; ** *p* < 0.01; *** *p* < 0.001 (Scheffe’s test) for comparisons of the treated and untreated LPS-stimulated DCs. (**B**) The inhibition percentage (%) of the subfractions, with and without genistein, of cytokine production was derived from the data in (A).

**Figure 5 foods-08-00543-f005:**
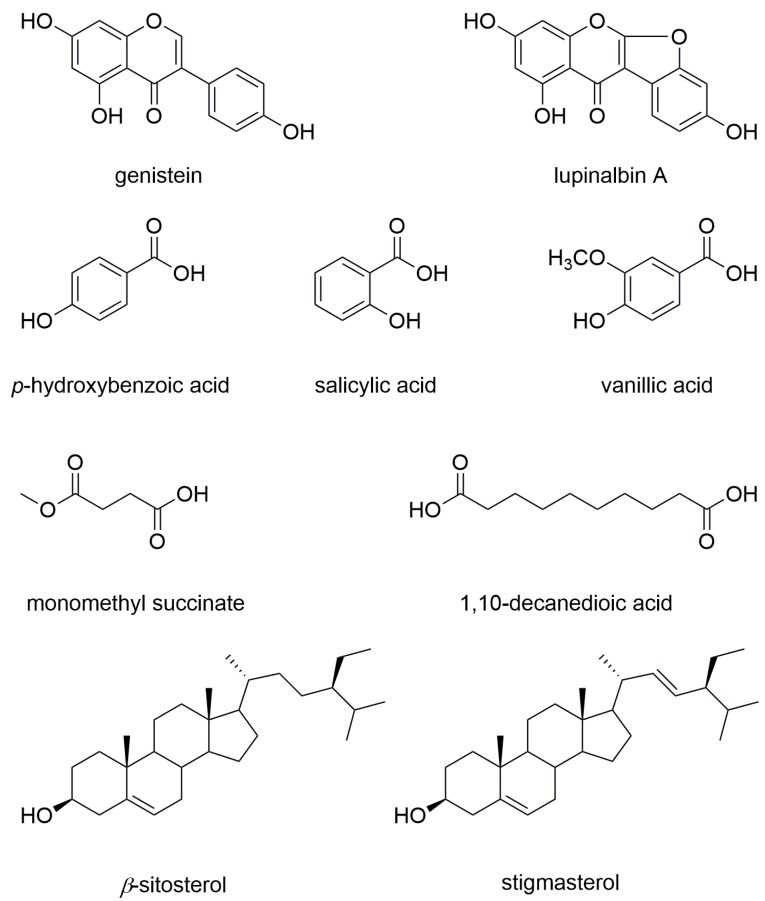
Chemical structures of the compounds identified in this study.

**Figure 6 foods-08-00543-f006:**
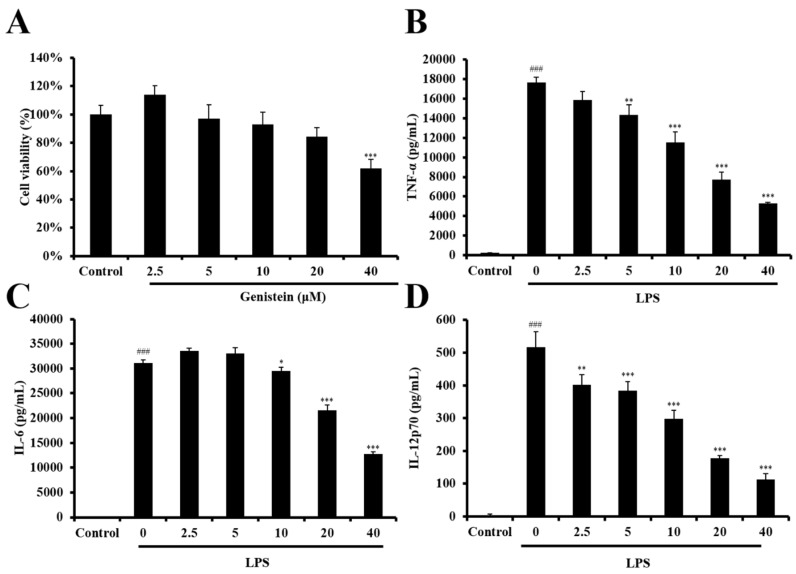
The effect of genistein on cell viability (A) and the production of pro-inflammatory cytokines in LPS-stimulated DCs (B, C, and D). (**A**) DCs were treated with genistein at various concentrations for 24 hours, and then their viability was measured by the CCK-8 assay (Sigma). DCs were untreated or treated with LPS (100 ng/mL) and LPS + genistein (2.5, 5, 10, 20, and 40 μM), as indicated. Supernatants were collected 6 h after treatment. The production of the cytokines TNF-*α* (**B**), IL-6 (**C**), and IL-12p70 (**D**) was measured by ELISA. The data shown are the mean ± SD of three independent experiments. ^###^
*p* < 0.001; * *p* < 0.05; ** *p* < 0.01; *** *p* < 0.001 (Scheffe’s test) for comparisons of the genistein-treated and untreated LPS-stimulated DCs.

**Figure 7 foods-08-00543-f007:**
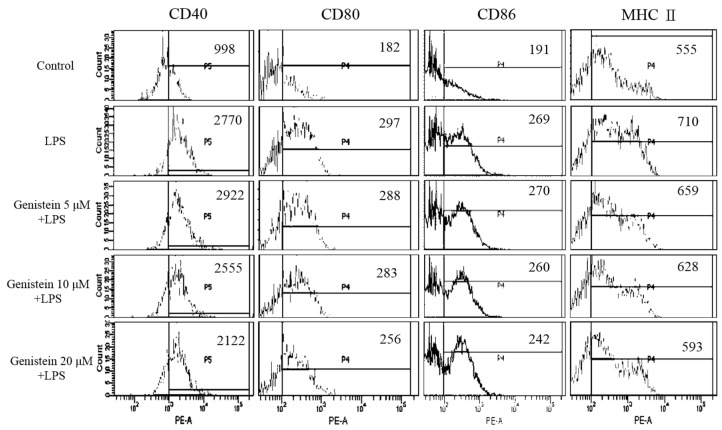
Genistein-mediated suppression of a maturation-associated surface marker on LPS-stimulated DCs. DCs were untreated or treated with LPS or LPS + genistein for 24 h. The suppression of major histocompatibility complex (MHC) class II and of co-stimulatory molecules (CD40, CD80, and CD86) was analyzed by flow cytometry. All of the data shown were gated on CD11c^+^ cells. All results are representative of three independent experiments.

**Table 1 foods-08-00543-t001:** Chemical shifts (*δ*_H_) and assignment of major compounds present in the D-4 and D-5 subfractions.

Compounds	*δ*_H_ (mult, *J* in Hz)
Genistein	13.01 (s), 8.13 (s), 7.44 (d, 8.7), 6.88–6.94 ^1^, 6.40 (d, 2.2), 6.28 (d, 2.2)
*p*-Hydroxybenzoic acid ^2^	7.90 (d, 8.9), 6.88–6.94 ^1^
Salicylic acid	7.88 (dd, 7.6, 1.7), 7.55 (ddd, 8.9, 7.6. 1.8), 6.88–6.94 ^1^
Vanillic acid	7.58 (dd, 8.2, 2.0), 6.88–6.94 ^1^, 3.89 (s)

^1^ Overlapping peaks at 6.88–6.94 ppm. ^2^ Only in the D-4 subfraction.

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
