# Peer review of "Bioactivity-Guided Fractionation and NMR-Based Identification of the Immunomodulatory Isoflavone from the Roots of Uraria crinita (L.) Desv. ex DC"

_foods, 2019, doi:10.3390/foods8110543_

Round 1

Reviewer 1 Report

Ping-Chen et al. attempted to study the Bioactivity-guided fractionation and NMR-based 3 identification of the immunomodulatory isoflavone 4 from the roots of Uraria crinita (L.) Desv. ex DC. Though this study is of interest and has merit, the authors need substantial improvement in the manuscript before being accepted in Foods Journal. So I recommend major revision.

Title

Title should be changed, because its doesn’t mention about inflammatory action of genestein.

Introduction

Please update the introduction with recent references.

Materials and methods

How did you do NMR analysis without performing HPLC analysis in order to check purity of the fraction? If you have already performed HPLC, please provide spectrum. Authors given a line about HPLC analysis in method sections. But there is no details about how HPLC analysis was performed? Such as, HPLC analysis conditions, column, solvent system, flow rate etc… Author mentioned they performed Flow cytometry analsyis for cell sorting, but in method sections there is information about this. How experiment was performed? NMR machine specification details are missing For performing column chromatography, author should mention solvent system ratio rather than simple writing hexane: EtoAC and Methanol solvent system.

Overall materials method section should be rewritten with updated points and proper reference.

Results and Discussion

How did you identified major constituents from UCME only by NMR analysis? Without PERFORMING purity analysis of sub-fraction (D1-D6) NMR data can’t interpret precisely. Though denoting literature reference, it should be more convinced by more analysis. If possible further analysis should be performed other spectroscopic analysis, such as EI-MS, FT-IR, HPLC-MS etc Why authors didn’t written the manuscript elaborately. I am not sure whether journal has page limitation or not. If not should be written clearly about that results and discussion part.

Author Response

Ping-Chen et al. attempted to study the Bioactivity-guided fractionation and NMR-based 3 identification of the immunomodulatory isoflavone 4 from the roots of Uraria crinita (L.) Desv. ex DC. Though this study is of interest and has merit, the authors need substantial improvement in the manuscript before being accepted in Foods Journal. So I recommend major revision.

The authors appreciate your suggestions that help improve our manuscript. Following are the reviewer comments with our response in blue.

Title: should be changed, because its doesn’t mention about inflammatory action of genestein.

Response: The activity examined in this study is the inhibitory activity of dendritic cell activation which is a kind of immunomodulation; thus, the immunomodulatory was used.

Introduction: Please update the introduction with recent references.

Response: Introduction was restructured, and some references were updated.

Materials and methods:

Point 1: How did you do NMR analysis without performing HPLC analysis in order to check purity of the fraction? If you have already performed HPLC, please provide spectrum.

Response 1: The NMR-based metabolomic approach has been applied for identification and quantification of metabolites in crude samples. This approach can provide a powerful strategy, allowing the direct study on crude extracts, and simultaneously identifying a wide range of (primary and secondary) metabolites, without unnecessary isolation procedures (Graziani et al., 2018 and Husin et al., 2019). HPLC chromatogram of D-4 and D-5 subfractions was supplemented as Figure S4.

Point 2: Authors given a line about HPLC analysis in method sections. But there is no details about how HPLC analysis was performed? Such as, HPLC analysis conditions, column, solvent system, flow rate etc…

Response 2: HPLC conditions used for D-4 and D-5 subfractions were added.

Point 3: Author mentioned they performed Flow cytometry analsyis for cell sorting, but in method sections there is information about this. How experiment was performed?

Response 3: Flow cytometry analysis information was added.

Point 4: NMR machine specification details are missing

Response 4: NMR machine specification was added.

Point 5: For performing column chromatography, author should mention solvent system ratio rather than simple writing hexane: EtoAC and Methanol solvent system.

Response 5: The isolation procedures was revised.

Point 6: Overall materials method section should be rewritten with updated points and proper reference.

Response 6: Material and Method section was rewritten according to reviewer’s comments.

Results and Discussion:

Point 1: How did you identified major constituents from UCME only by NMR analysis? Without PERFORMING purity analysis of sub-fraction (D1-D6) NMR data can’t interpret precisely. Though denoting literature reference, it should be more convinced by more analysis. If possible further analysis should be performed other spectroscopic analysis, such as EI-MS, FT-IR, HPLC-MS etc

Response 1: The 1H NMR signal intensity is absolutely proportional to the abundance of hydrogens. It could provide useful information about the relative abundance of the different metabolites by integrating the 1H NMR resonances (Lopatriello et al., 2017 and Mediani et al., 2017).

Thus, HPLC together with other spectroscopic analysis is not necessary for determination of the major metabolites.

Point 2: Why authors didn’t written the manuscript elaborately. I am not sure whether journal has page limitation or not. If not should be written clearly about that results and discussion part.

Response 2: Results and Discussion section was revised according to reviewer’s comments. Thank you again.

Reviewer 2 Report

Comments:

The manuscript described identification of genistein from Uraria crinita as the immunomodulatory constitute. However, it has been reported that genistein is the major constitute of Uraria crinite (Wang Y.Y. et al., Studies on chemical constituents in roots of Uraria crinite,  Journal of Chinese Pharmaceutical Sciences, 2009; 44(16):1217-1220.). In addition, genistein itself has been reported to possess potent mmunomodulatory activity (Kim et al., genistein inhibits proinflammatory cytokines in human mast cell activation through the inhibition of the ERK pathway, International Journal of Molecular Medicine, 2014; 34(6):1669-1674.; Dijsselbloem et al., A critical role for p53 in the control of Nf-kB-dependent gene expression in TLR4-stimulated dendritic cells exposed to genistein. Journal of Immunology; 2007, 178(8), 5048-5057.). Furthermore, Yen et al. has reported that genistein from Uraria crinita was the major immunomodulatory constitute (Yen GC, Lai HH, and Chou HY Nitric oxide-scavenging and antioxidant effects of Uraria crinita root. Food Chemistry, 2001; 74(4):471-478.). The present study used a different method (NMR-based) to repeat previous one and did not find any different results. Thus, the reviewer did not support the acceptance of the manuscript.

Author Response

The manuscript described identification of genistein from Uraria crinita as the immunomodulatory constitute. However, it has been reported that genistein is the major constitute of Uraria crinite (Wang Y.Y. et al., Studies on chemical constituents in roots of Uraria crinite, Journal of Chinese Pharmaceutical Sciences, 2009; 44(16):1217-1220.).

In addition, genistein itself has been reported to possess potent mmunomodulatory activity (Kim et al., genistein inhibits proinflammatory cytokines in human mast cell activation through the inhibition of the ERK pathway, International Journal of Molecular Medicine, 2014; 34(6):1669-1674.; Dijsselbloem et al., A critical role for p53 in the control of Nf-kB-dependent gene expression in TLR4-stimulated dendritic cells exposed to genistein. Journal of Immunology; 2007, 178(8), 5048-5057.). Furthermore, Yen et al. has reported that genistein from Uraria crinita was the major immunomodulatory constitute (Yen GC, Lai HH, and Chou HY Nitric oxide-scavenging and antioxidant effects of Uraria crinita root. Food Chemistry, 2001; 74(4):471-478.). The present study used a different method (NMR-based) to repeat previous one and did not find any different results. Thus, the reviewer did not support the acceptance of the manuscript.

Response: The authors are thankful for reviewer’s suggestions. As to the reviewer’s mentioned, total four previous studies related to this research on Uraria crinita were demonstrated as major reasons for rejecting our manuscript. Those previous studies have covered some specific topics including the investigation on chemical constituents of Uraria crinita (Wang et al., 2009) and the immunomodulatory activity of genistein (Kim et al., 2014; Dijsselbloem et al., 2007; Yen et al., 2001). Genistein was indeed isolated from this plant previously. It has been reported to show potent immunomodulatory activity. However, the correlations between genistein and the immunomodulatory activity of this plant have not been systematically concluded and proved. Otherwise, all the herbs containing genistein could be supposed to show potent inflammatory activity. In addition, whether genistein plays a critical role in the immunomodulatory activity of Uraria crinita was still unknown. Actually, Wang’s and Yen’s reports did not provide strong and direct evidences to support the bioactivity of genistein in Uraria crinita, if reviewer checked their preliminary results. Wang reported the chemical constituents, but there is no biological assessment on the isolates. Yen reported the antioxidant and nitric oxide-scavenging activities of the Uraria crinita roots methanol extracts and identified genistein as one of the major components. The contribution of genistein in antioxidant effects of the Uraria crinita roots methanol extracts was not determined. Thus, we disagree with reviewer’s comments that there is no new finding in our work. 

An extensively investigated compound genistein was identified from the combination of the bioactivity-guided isolation of UCME and NMR-based identification. Nevertheless, the correlation between genistein and the immunomodulatory activity of this plant was solidly confirmed by the results of the genistein-knockout subfractions showing the much less inhibitory activity by our group. Also, the critical role was also supported by the other isolates with poor inhibitory activity. On the basis of the above information, the presence of genistein in fraction D (major in D-4 and D-5 subfractions) was directly linked to the immunomodulatory activity of this plant.

NMR-based metabolomics approach has not been widely used for rapid identification of bioactive compounds in partially purified subfractions. But, this is powerful and can prevent time-consuming isolation procedures, providing a rapid identification of bioactive metabolites. The importance in this study is to elucidate a rapid and precise tool for clarifying the central role of genistein in modulating DCs activation. Also, the combination of bioactivity-guided isolation and NMR-based identification can be applied for further investigations on herbal products including functional foods and herbal medicines.

Round 2

Reviewer 1 Report

All corrections were made.

Author Response

Response: We would like to thank you for the positive feedback and helpful comments for improving the manuscript. According to editor’s suggestions, our manuscript has been submitted for MDPI English language editing service. Thank you for your time.

Reviewer 2 Report

In the first round of the review, the reviewer comment that both identification of genistin in the plant and identification of genistein as a potent immunomodulaor are not new information. To the author's replay "The contribution of genistein in antioxidant effects of the Uraria crinita roots methanol extracts was not determined. Thus, we disagree with reviewer’s comments that there is no new finding in our work.", the reviewer thinks the reply is not valuable enough to scientist readers.

Author Response

Our manuscript was revised according to the reviewers’ suggestions. Thank you for your comments that will help improve our manuscript. After reading the previous studies carefully, we explain the importance and novelty of this work as below:

First, Wang (2009) reported chemical constituents in roots of Uraria crinita. (The reference was added in revision version.) Among the isolates, genistein is not the major constitute (14 mg of genistein was obtained from 10 kg dry material, not more than other isolates). In contrast, we isolated 24 mg of genistein from 1.136 kg dry material by target isolation of the immunomodulatory metabolites. The yield of genistein was fifteen times more than that from previous study. Moreover, Yen (2001) reported the antioxidant and nitric oxide-scavenging activities of the crude methanol extracts and the EtOAc-soluble fraction but not for genistein. Also, Yen (2001) determined genistein as the major compound and active compound by HPLC with a diode array detector. (In our study, the relative abundance could be measured by integrating 1H NMR signals.) In fact, the major peak observed in a chromatogram using a diode array detector set at 260 nm only suggested the absorbance of that compound at 260 nm is stronger than others. The absorbance of the organic compounds is correlated not only to concentration but to the molar attenuation coefficient. (Flavonoids often have strong absorption in a chromatogram using a diode array detector.) There is no evidence that genistein is a major constitute in previous study. The contribution of that to bioactivity needed further elucidation.

Second, the immunomodulatory effects of UC roots on dendritic cells have not been reported (Line 87). Even though genistein has been reported to possess potent immunomodulatory activity in previous studies (Kim et al., 2014; Dijsselbloem et al., 2007; Yen et al., 2001), there is no report revealing the direct association between the immunomodulatory activity of Uraria crinita and genistein. Whether genistein could be a chemical marker of UC is still unknown. Also, it is really common that the phytochemical investigations from different resources led to the same constituents. In herbal products, the active ingredient may require cofactors to achieve therapeutic effects. Without being proved, the association between genistein and the immunomodulatory activity of Uraria crinita could not be concluded.

There are few reports about both NMR-based metabolomics approach (for identification of metabolites from bioactive fractions) and compound-knockout isolation (for confirming the bioactivity contribution of that). We combined these two for elucidating the central role of genistein in modulating DCs function of UC roots. The idea is valuable to readers who have an interest in discovering the active ingredient from herbal products.

In addition to the identification of genistein, the fractionation and HPLC isolation of UCME gave 8 compounds, including an isoflavone (lupinalbin A), three phenolic acids (p-hydroxybenzoic acid, salicylic acid, and vanillic acid), two fatty acids (monomethyl succinate and 1,10-decanedioic acid), and two steroids (a mixture of β-sitosterol and stigmasterol). Among the isolates, lupinalbin A, p-hydroxybenzoic acid, vanillic acid, monomethyl succinate, and monomethyl succinate were isolated from this herb for the first time. However, isolated compounds, especially from immunomodulatory subfractions (D-4 and D-5), did not show potent inhibitory activity. (p.7) Furthermore, the 1H NMR profiling of the bioactive subfractions revealed the presence of other unknown genistein derivatives, which may contribute slightly to the immunomodulatory effects for their low abundance or poor activity. The well-known bioactive isoflavonoids, including daidzein, formononetin, equol, and glycitein, were not present as major metabolites in the fraction D. (p.8) Our study showed the evidence proving that genistein plays a critical role in the immunomodulatory activity of UC.

The variable constituents in herbal products owing to genetic, cultural and environmental factors have made the quality of herbal medicines difficult to control. This work could provide a precise strategy for discovering the active ingredient(s) for quality control of herbal products.

Overall, the findings indicated that UC roots can potentially be applied as an immunosuppressive functional food of which genistein can be a chemical marker for quality control. The importance in this study is not the isolation of genistein alone. We did focus on the strategy mentioned above for identification of genistein as a chemical marker from this functional food but not determine it as a potent immunomodulator. The identification of genistein is only part of overall elucidation for the immunomodulatory activity of Uraria crinita.

Thank you for your time.